# Genome- and Transcriptome-Wide Association Studies Identify Susceptibility Genes and Pathways for Periodontitis

**DOI:** 10.3390/cells12010070

**Published:** 2022-12-23

**Authors:** Guirong Zhu, Xing Cui, Liwen Fan, Yongchu Pan, Lin Wang

**Affiliations:** 1Department of Orthodontics, The Affiliated Stomatology Hospital of Nanjing Medical University, Nanjing 210029, China; 2Jiangsu Province Key Laboratory of Oral Diseases, Nanjing Medical University, Nanjing 210029, China; 3Jiangsu Province Engineering Research Center of Stomatological Translational Medicine, Nanjing Medical University, Nanjing 210029, China; 4State Key Laboratory of Reproductive Medicine, Nanjing Medical University, Nanjing 210029, China

**Keywords:** genome-wide association study (GWAS), transcriptome-wide association study (TWAS), periodontitis

## Abstract

Several genes associated with periodontitis have been identified through genome-wide association studies (GWAS); however, known genes only explain a minority of the estimated heritability. We aimed to explore more susceptibility genes and the underlying mechanisms of periodontitis. Firstly, a genome-wide meta-analysis of 38,532 patients and 316,185 healthy controls was performed. Then, cross- and single-tissue transcriptome-wide association studies (TWAS) were conducted based on GWAS summary statistics and the Genotype-Tissue Expression (GTEx) project. Risk genes were evaluated to determine if they were differentially expressed in periodontitis sites compared with unaffected sites using public datasets. Finally, gene co-expression network analysis was conducted to identify the functional biology of the susceptible genes. A total of eight single nucleotide polymorphisms (SNPs) within the introns of lncRNA *LINC02141* approached genome-wide significance after meta-analysis. *EZH1* was identified as a novel susceptibility gene for periodontitis by TWAS and was significantly upregulated in periodontitis-affected gingival tissues. *EZH1* co-expression genes were greatly enriched in the cell-substrate junction, focal adhesion and other important pathways. Our findings may offer a fundamental clue for comprehending the genetic mechanisms of periodontitis.

## 1. Introduction

Periodontitis is a multifactorial disease with bacteria, genetic and environmental factors playing etiologic roles [1]. It is characterized by the loss of periodontal tissue support manifested by clinical attachment loss (CAL) and radiographically assessed alveolar bone loss, presence of periodontal pocketing and gingival bleeding [2,3]. Periodontitis is initiated and sustained by microbial infection in subgingival dental biofilm by periodontal pathogens including *Porphyromonas gingivalis*, *Tannerella forsythia*, and so on [4]. The innate and adaptive immune responses against these periodontal bacteria account for most of the periodontitis-related tissue damage [5].

Genetic factors considerably contributed to the development of periodontitis. Periodontitis had approximately 50% heritability after adjusting for behavioral variables in a twin study, suggesting a substantial genetic contribution to the etiology of periodontitis [6]. A systematic review revealed that twenty-nine per cent of the variance of periodontitis traits is related to genetic factors based on twin and family studies [7]. Variants in at least 65 genes, especially genes involved in inflammation, have been implicated to be associated with periodontitis [8,9]. Besides the genetic factors, the impacts of environmental risk have been acknowledged for decades. Tobacco use is one of the most significant avoidable risk factors for the occurrence and development of periodontal disease [10]. Moreover, medicines, poverty, obesity, uncontrolled diabetes and mental stress are also strongly associated with periodontitis [11]. Thus, the severity of periodontitis is determined by genetic and environmental factors [12].

In recent years, genome-wide association studies (GWAS) have identified several loci in the genome linked to periodontitis. According to the GWAS catalog, nineteen GWAS studies on periodontitis have been conducted [13]. A total of six variants (rs1537415, *GLT6D1*; rs242016, *EFCAB4B*; rs729876, *SHISA9*; rs11084095, *SIGLEC5*; rs4284742, *SIGLEC5*; and rs2738058, *DEFA1A3*) meet the significant threshold (*p* < 5 ×10^−8^) [14,15,16,17].

Although GWASs have achieved success in identifying associated loci, the statistical power and biological interpretation of GWAS results remain to be addressed [18]. Meta-analysis, gene-gene and gene-environment interactions analysis, larger samples, wider computing and some other methods are used to improve statistical power [18]. Many GWAS hits are non-coding variants residing in intergenic or intronic regions, and little is known about their biological significance. Furthermore, strong linkage disequilibrium (LD) within gene-content haplotypes makes it difficult to identify candidate variants and genes [19,20]. To overcome this issue, it is essential to translate GWAS findings into a better understanding of the biology underlying periodontitis risk by integrating other methods. 

The Genotype-Tissue Expression (GTEx) database of expression quantitative trait loci (eQTL), a tissue bank to investigate the correlation between single nucleotide polymorphisms (SNPs) and gene expression in human tissues, serves to fill the gap in variants, genes, and complex traits [21]. Transcriptome-wide association study (TWAS), an integrated analysis of eQTL mapping with GWAS, can be applied to explore the most likely target genes and reveal the regulatory mechanisms underlying complex traits [22]. Single-tissue TWAS methods such as MetaXcan [23], PrediXcan [24], and FUSION [25] are widely used for diseases or traits with relevant tissues. Considering a certain number of relevant tissues were scarce and the similarity in transcription regulation across multiple tissues, researchers created a cross-tissue analysis method called UTMOST (Unified Test for MOlecular SignaTures) [20,26]. Cristina et al. used UTMOST to analyze the largest autism spectrum disorder (ASD) GWAS summary statistics and found that *NKX2-2* and *BLK* were associated with ASD but not restricted to the brain tissue [27]. Using UTMOST and FUSION in combination, Zhu et al. identified two novel susceptibility genes (*DCAF16* and *CBL*) for lung cancer [28].

In this study, we aimed to explore new genes associated with periodontitis and discover the biological mechanism underlying the association. A meta-analysis was performed to identify variants associated with periodontitis in Europeans using two large-scale GWAS datasets. We then conducted a TWAS analysis using both the UTMOST and FUSION methods to explore more susceptibility genes. Potential genes were validated using two gene expression datasets. Finally, downstream pathway enrichment analysis was conducted to confirm the function of susceptible genes in the etiology of periodontitis.

## 2. Materials and Methods

### 2.1. Periodontitis GWAS Data

Summary statistics for periodontitis from the Gene-Lifestyle Interactions in Dental Endpoints (GLIDE) consortium (https://data.bris.ac.uk/data/dataset/2j2rqgzedxlq02oqbb4vmycnc2, accessed on 16 June 2022) and FinnGen (https://www.finngen.fi/en/access_results, accessed on 16 June 2022) were downloaded (Appendix A) [29,30]. GLIDE consists of systematic meta-analysis statistics based on nine studies with 17,353 periodontitis cases and 28,210 controls from Europe ranging from 17 to 93 years, the specifics of which have been described before [29]. FinnGen comprises 21,179 periodontitis cases and 287,975 controls with a median age of 63 years of European descent. Additional information on recruitment, genotyping, and quality control was previously provided [30]. Before analysis, LiftOver altered the FinnGen location (hg38) to match the location of GLIDE (hg19) (https://genome.sph.umich.edu/wiki/LiftOver, accessed on 21 October 2022). Then, results from GLIDE and FinnGen were incorporated into a fixed effect inverse-variance weighted meta-analysis using Metal software, with a classical approach using effect size estimates and standard errors [31]. The Manhattan and quantile-quantile plots were generated using the R package “qqman”. LocusZoom was used to visualize the genomic regions of significance (http://locuszoom.sph.umich.edu/, accessed on 21 October 2022) [32].

### 2.2. Cross-Tissue TWAS Analysis Using UTMOST

UTMOST integrated GWAS summary statistics and gene expression data of 44 tissues from 450 individuals in GTEx V7 to execute a cross-tissue association test [20]. First, we downloaded UTMOST pre-calculated covariance matrices (https://github.com/Joker-Jerome/UTMOST, accessed on 21 October 2022). Single-tissue association tests for 44 tissues were then performed by integrating GWAS periodontitis summary statistics and genetic expression weights. Finally, the relationships between 17,290 genes and periodontitis in 44 tissues were calculated by a joint GBJ test in combination with single-tissue gene-trait association results. The transcriptome-wide significance for the joint test was set as *p*-value < 1 × 10^–4^. 

### 2.3. Single-Tissue TWAS Analysis Using FUSION

In our study, a single-tissue TWAS analysis was performed using FUSION [25] to minimize false positive errors and train independent imputation models for various tissues. The precomputed predictive models for tissues in GTEx V7 were downloaded from the official FUSION website (http://gusevlab.org/projects/fusion/, accessed on 21 October 2022). We then estimated the association of each gene with periodontitis, taking precomputed gene expression weights together with GWAS summary statistics. LD references were derived for Europeans from the 1000 Genome Project. We selected significant UTMOST genes with *P*_FUSION_ < 0.05 in at least five tissues for further analysis.

### 2.4. Expression Analysis of Candidate Genes in the Gingival Tissues of Periodontitis Patients

GSE10334 [33] and GSE16134 [34] (https://www.ncbi.nlm.nih.gov/geo/, accessed on 21 October 2022) provided gene expression signatures in healthy and diseased gingival tissues. We queried these two datasets to verify whether candidate genes were differentially expressed in periodontitis sites compared with unaffected sites. In GSE10334, 90 non-smokers with periodontitis provided a total of 247 tissue samples (64 from healthy sites and 183 from periodontitis). RNA was extracted, amplified, reverse-transcribed, labeled, and hybridized using Affymetrix Human Genome U133 Plus 2.0 arrays. GSE16134 contained 120 systemically healthy non-smokers with periodontitis, including 310 gingival tissue samples (241 samples from affected and 69 from unaffected sites). Differentially expressed genes (DEGs) between periodontitis and healthy sites were screened using the interactive web tool GEO2R (http://www.ncbi.nlm.nih.gov/geo/geo2r/, accessed on 21 October 2022) with *P*_adj_ < 0.05. The plots were plotted using GraphPad Prism software (GraphPad Software, San Diego, CA, USA).

### 2.5. Gene Annotation and Enrichment Analyses

We queried the Comparative Toxicogenomics Database (CTD; http://ctdbase.org/, accessed on 21 October 2022), which seeks to clarify links between chemicals, genes, and diseases as well as enhance the understanding of how environmental exposures affect human health [35]. Furthermore, considering that genes with comparable expression patterns across a set of samples may have an intrinsic and functional relationship, we quantified genes co-expressed with the causal genes to determine and understand their biological functions and mechanisms. The correlations were determined using 310 tissues from 120 samples in GSE16134 by Pearson correlation analysis. Statistically significant co-expressed genes with *p*-values satisfying the Bonferroni correction threshold of 9.14 × 10^–7^ (0.05/54,676) were included in the Gene Ontology (GO) and Kyoto Encyclopedia of Genes and Genomes (KEGG) enrichment analyses using “clusterProfiler” from the R package [36].

## 3. Results

### 3.1. Novel Risk Locus Discovered by Meta-Analysis

A flowchart of the study is displayed in Appendix A. GLIDE and FinnGen included 13,284,112 and 16,962,023 SNPs, respectively. A total of 4,467,917 SNPs were matched on the chromosome, nucleotide position, alleles, and direction of the log odds ratios (OR) in 38,532 periodontitis cases and 316,185 controls. Quantile-quantile plots did not show any evidence of a substantial inflation rate, ruling out a substantial cryptic population substructure and differing genotypic variants between the periodontitis cases and controls (λ_GLIDE_ = 1.00, λ_FINNGEN_ = 1.08, λ_Meta_ = 1.05, Appendix A). 

Neither GLIDE nor FinnGen SNPs exhibited any significant evidence of an association with periodontitis (*p* < 5 × 10^−8^, Figure 1a,b). After meta-analysis, eight SNPs mapped to 16q21 physically (rs11076352, rs28635813, rs28394006, rs11864657, rs2406892, rs12325580, rs12922928, and rs2216750) passed the genome-wide significance threshold after adjusting for multiple testing (*p* < 5 × 10^−8^, Figure 1c, Appendix A). All of these SNPs were non-coding variants situated in introns of lncRNA *LINC02141* and were in high LD (Figure 2a). Figure 2b presents their estimated OR, 95% confidence interval (CI), and physical positions.

### 3.2. Cross-Tissue Transcriptome-Wide Significant Genes

To further explore the associations between gene expression levels and periodontitis, we integrated our large-scale GWAS summary statistics with the gene expression matrix of 44 different tissues from 450 participants from the GTEx project using UTMOST. Since other genes except *SIGLEC14* did not pass the Bonferroni correction, we set the threshold as 1 × 10^–4^ in the study to minimize false negative errors. Six genes (*SIGLEC14*, *SAMD9L*, *EZH1*, *MRPS23*, *SIGLEC5*, and *TMED7*) reached statistical significance (Figure 3a, Table 1 and Appendix A).

### 3.3. Single-Tissue Transcriptome-Wide Significant Genes

To better characterize the six candidate genes, we separately evaluated their associations in every tissue using FUSION. Among them, four genes (*SIGLEC14*, *EZH1*, *MRPS23*, and *SIGLEC5*) were significant in at least five tissues, with *P*_FUSION_ < 0.05 (Figure 3b, Table 1, Appendix A). Interestingly, *EZH1* and *MRPS23* were newly identified, whereas previous studies have verified *SIGLEC14* and *SIGLEC5* as genetic risk factors for periodontitis [37]. However, *SAMD9L* was not significant in any tissue and *TMED7* was only significant in brain cerebellar hemisphere tissue.

### 3.4. Significant Upregulation of EZH1 in Periodontitis

TWAS is widely used to infer disease-related genes based on SNP genotypes. Therefore, if the genes identified by TWAS are relevant, their expression should be dysregulated in periodontitis. Because the whole-genome microarrays did not include *SIGLEC14*, its differential expression was impossible to evaluate according to GSE10334 and GSE16134. *MRPS23* and *SIGLEC5* were upregulated in periodontitis-affected sites of gingival tissue in GSE10334 compared with unaffected sites, but there were no differences in GSE16134 (Figure 4b,c,e,f). Notably, the expression of the newly identified gene *EZH1* increased in periodontitis sites in both GEO datasets (Figure 4a,d). 

### 3.5. Functional Annotation of EZH1

As the only differentially expressed gene in both GEO datasets, we performed a functional annotation on *EZH1.* According to the CTD, some diseases such as inflammation and necrosis were associated with *EZH1* (Figure 5a). Among them, inflammation was inferred to have the top score of 140.01 through interaction with 41 chemicals including smoke, soot and so on (Figure 5a,b, Appendix A). To address the pathogenesis of *EZH1* in periodontitis, we also explored the genes co-expressed with *EZH1*. The expression levels of 4724 genes were significantly correlated with *EZH1* after Bonferroni correction using the expression data in GSE16134 owing to its larger sample size. Robust statistical evidence for GO enrichment was observed for cell-substrate junctions (GO: 0030055, *P*_adjust_ =1.04 × 10^−11^) and focal adhesion (GO: 0005925, *P*_adjust_ =1.04 × 10^−11^) (Figure 5c, Appendix A). The KEGG analysis revealed that genes co-expressed with *EZH1* were closely associated with Epstein-Barr virus (EBV) infection (KEGG: hsa05169, *P*_adjust_ = 9.93 × 10^−7^), lysosome (KEGG: hsa04142, *P*_adjust_ = 9.93 × 10^−7^), and protein processing in the endoplasmic reticulum (KEGG: hsa04141, *P*_adjust_ = 9.93 × 10^−7^), all of which are associated with periodontitis (Figure 5d, Appendix A).

## 4. Discussion

In this study, we first meta-analyzed two large GWASs to detect periodontitis-associated SNPs. Eight SNPs within the introns of lncRNA *LINC02141* were associated with periodontitis. UTMOST then served as a cross-tissue TWAS method and identified six genes associated with periodontitis, four of which were validated by FUSION. Furthermore, *EZH1* was significantly upregulated at periodontitis sites. Genes co-expressed with *EZH1* were considerably enriched in biological processes such as cell-substrate junctions and focal adhesion. In summary, we identified eight SNPs associated with periodontitis and highlighted the possible mechanisms of *EZH1*.

In 16q21, there were eight SNPs within the introns of lncRNA *LINC02141* with *p* < 5 × 10^−8^. LncRNAs (e.g., *AWPPH* and *FGD5-AS1*) were reported to offer a strong foundation for periodontitis diagnosis and prognosis [38,39,40]. Unfortunately, *LINC02141* was not detected by GSE10334 and GSE16134, making it difficult to study its expression level. Motif analyses using HaploReg v4.1 [41] (http://pubs.broadinstitute.org/mammals/haploreg/haploreg.php, accessed on 21 October 2022) revealed that the A allele of rs11076352, the most significant SNP, altered the affinity of GATA (score: −6.9 to 5). GATA has repeatedly been reported as a transcription factor associated with smoking and periodontitis [42,43]. Although associations between the SNPs within *LINC02141* and periodontitis were observed, their specific molecular mechanisms need to be elucidated in further studies.

The integration of GWAS and eQTL data has been successfully used to explore gene-trait associations. Li et al. used a Bayesian statistical method called Sherlock to systematically incorporate GWAS and eQTL data for periodontitis from whole blood. They identified ten significant genes, including *SIGLEC5* and *SIGLEC14* [44]. In our study, UTMOST was applied to develop imputation models for 44 tissues, considering the substantial shared gene regulation across tissues and compensating for the lack of adequate gingival tissue data. To decrease false-positive results, FUSION assisted in the evaluation of the associations in different tissues. Finally, we identified two previously reported genes (*SIGLEC5*, *SIGLEC14*) and two novel genes (*EZH1*, *MRPS23*), proving that TWAS is a supplemental strategy for studying periodontitis’ etiology. 

Further differential expression analysis revealed that *EZH1* was consistently upregulated in periodontitis sites among patients with periodontitis. *EZH1*, also known as an enhancer of the zeste one polycomb repressive complex 2 subunit, is a protein-coding gene. According to the CTD database, inflammation was associated with *EZH1* with the highest inference score. Smoke, soot and another 39 chemicals belonged to the inference network. It is well known that periodontitis is an inflammatory disease characterized by the destruction of the supporting tissues around the teeth. Meanwhile, Ezh1 is the most upregulated histone methyltransferase during dendritic cell maturation, silencing of which significantly suppressed Toll-like receptors and triggered production of cytokines, including IL-6, TNF-α, and IFN-β, in dendritic cells (DCs) and macrophages [45]. These cytokines are known as pro-inflammatory cytokines and have pleiotropic effects on lymphocyte promotion as well as tissue damage [46]. For example, IL-6 can participate in alveolar bone resorption and attachment loss supported by the modulation of the Th17 periodontal response [47]. Therefore, *EZH1* may participate in the development of periodontitis through inflammation.

Moreover, GO enrichment analysis indicated that genes co-expressed with *EZH1* were mainly involved in cell-substrate junctions and focal adhesion. Focal adhesion is also a kind of cell-substrate junction that anchors the cell to the extracellular matrix. When infected with *Porphyromonas gingivalis*, a major pathogen of periodontitis, both cell-matrix and cell-cell adhesion interfered in the oral keratinocyte cell line, which could compromise the integrity of the epithelia and result in the generation of gingival pockets between the cementum and periodontal epithelium [48]. Thus, genes correlated with *EZH1* may play a role in cell and extracellular matrix interactions and participate in the development of periodontitis. The KEGG pathway showed a strong correlation with the EBV infection, lysosomes, and protein processing in the endoplasmic reticulum (ER). EBV plays a crucial role in inflammatory cytokine production and osteoclast differentiation by interacting with oral cells or the macrophage lineage, which initiate and advance periodontitis [49]. Kenichi et al. provided evidence that EBV interacts with some periodontopathic bacteria and is intimately related to the beginning and progressive stages of chronic periodontitis [50]. Lysosomes can fuse with autophagosomes to form autolysosomes during autophagy [51]. *Porphyromonas gingivalis* in the cytosol is usually degraded by lysosomes, but *Porphyromonas gingivalis* co-localized with double-membrane vacuoles survives by impairing the formation of autolysosomes [52]. In addition, if cells are infected with pathogenic bacteria, expression of autophagy-related proteins and autophagosomes increases [53]. When periodontal disease develops, autophagy has an impact on immunological function, inflammation, and the homeostasis of alveolar bone [52]. Therefore, lysosomes, as an integral part of autophagy, could play an important role in periodontitis. When protein processing fails in the endoplasmic reticulum, the accumulation of misfolded proteins in the ER causes ER stress. ER stress could affect the development of periodontal diseases [54]. Altogether, the GO and KEGG enrichment analysis results provide additional evidence for the potential role of genes co-expressed with *EZH1* in the pathological process of periodontitis.

Although this study identified exciting genes and pathways, there are limitations worth mentioning. Even though a large sample size was included in the GWAS and two TWAS methods were used to reduce false positives, functional research is necessary to better reveal the biological mechanisms of periodontitis. Additionally, a lack of gingival-related cells or tissue is another limitation in our study, despite genetic correlation studies demonstrating similar expression regulation between different tissues [20]. In the future, available eQTL data from the gingiva will be of great benefit to validate our findings. 

## 5. Conclusions

In the present study, several SNPs reached genome-wide significance for periodontitis. Furthermore, TWAS and in silico analyses illustrated that *EZH1*, genes co-expressed with which were significantly enriched in cell-substrate junction and other important pathways, may contribute to the risk of periodontitis. Further research is required to corroborate our findings and to dissect the genetic architecture of periodontitis. 

## Figures and Tables

**Figure 1 cells-12-00070-f001:**
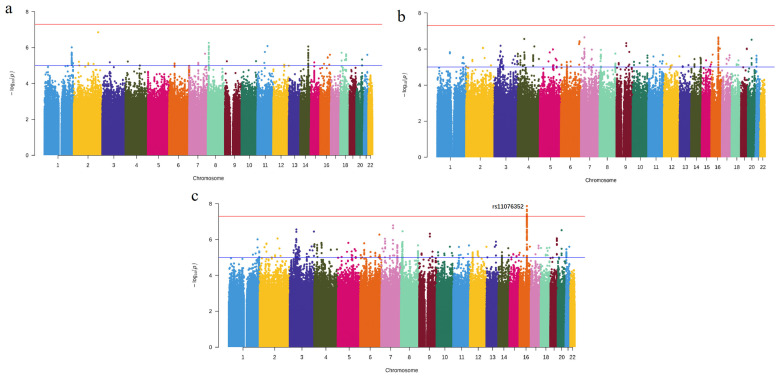
Manhattan plot of −log10 (*p*-values) of the GWAS results. (**a**) Summary statistics from the GLIDE consortium. (**b**) Summary statistics from FinnGen. (**c**) Meta-analysis results. The red and blue horizontal lines indicate the level of genome-wide significance (*p* = 5.0 × 10^−8^) and nominal significance (*p* = 1.0 × 10^−5^), respectively. GWAS, genome-wide association study.

**Figure 2 cells-12-00070-f002:**
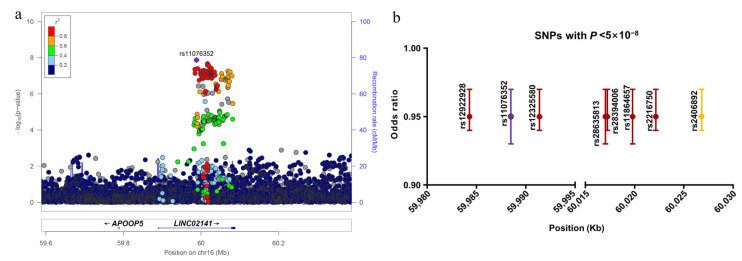
Regional plots and effective size of significant SNPs. (**a**) Regional plots of association results and recombination rates for the genome-wide significant SNP rs11076352. (**b**) OR and 95% CI under an additive model for SNPs with *p* < 5 × 10 ^−8^. SNPs, single nucleotide polymorphisms; OR, odds ratio; 95% CI, 95% confidence interval.

**Figure 3 cells-12-00070-f003:**
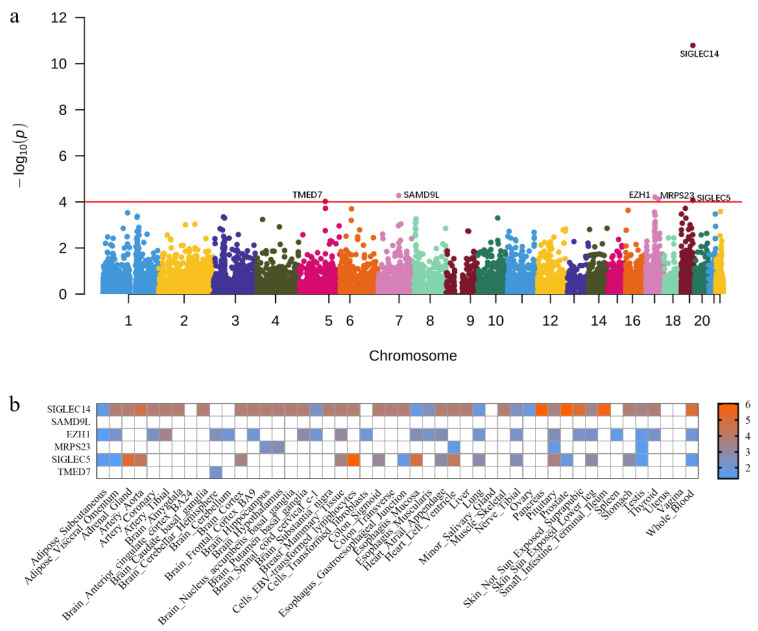
Transcriptome-wide association study results for periodontitis. (**a**) Manhattan plot of cross-tissue TWAS using UTMOST. The y-axis represents the *p*-value in −log10 scale. Genes with *p*-value < 1 × 10^−4^ are labeled. (**b**) The heatmaps of the six genes are segregated by tissues. The statistics represent the *p*-value of FUSION on a −log10 scale. The white lattice represents *P*_FUSION_ > 0.05 or unavailable statistics. TWAS, transcriptome-wide association study.

**Figure 4 cells-12-00070-f004:**
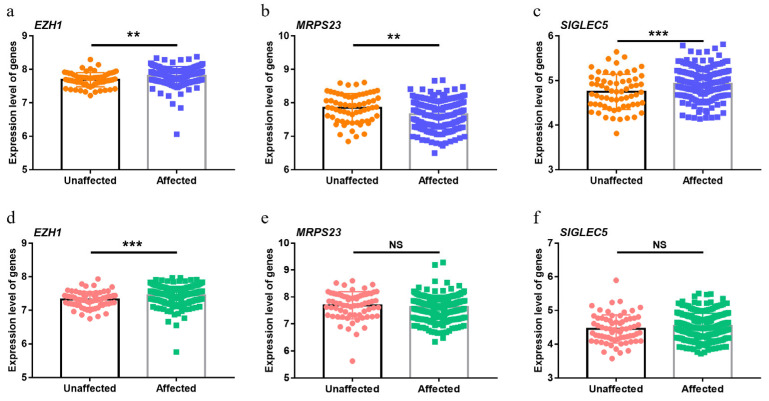
Dysregulation of top causal genes in periodontitis. Differentially expressed genes in the affected and unaffected gingival tissue sites among periodontitis patients. Data from GSE10334 (**a**–**c**) and GSE16134 (**d**–**f**). *EZH1* was significantly upregulated in the periodontal ligament in periodontitis sites compared with unaffected sites. NS, *p* > 0.05; ** *p* < 0.01; *** *p* < 0.001.

**Figure 5 cells-12-00070-f005:**
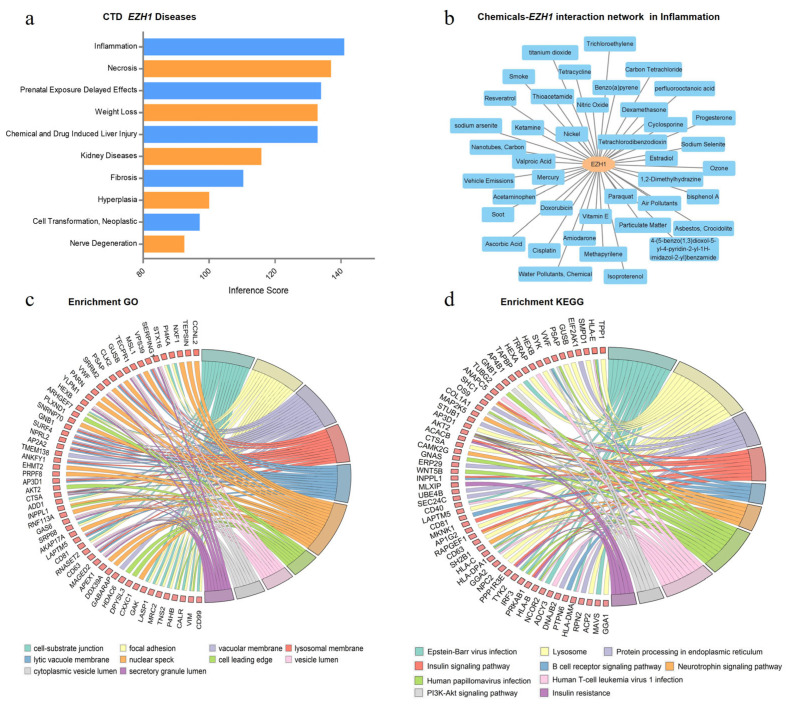
Functional annotation of *EZH1*. (**a**) Disease categories associated with *EZH1* gene in the CTD database with inference scores. (**b**) A total of 41 chemicals interacted with *EZH1* and correlated with inflammation. Circle diagrams of GO (**c**) and KEGG (**d**) enriched pathways for genes co-expressed with *EZH1* from GSE16134. CTD, the Comparative Toxicogenomics Database; GO, Gene Ontology; KEGG, Kyoto Encyclopedia of Genes and Genomes.

**Table 1 cells-12-00070-t001:** Significant genes in the TWAS analysis of periodontitis.

Gene	Chr	BP	*P* _UTMOST_	*P* _FUSION_	*n*	Reported
** *SIGLEC14* **	19	52145806	1.63 × 10^−11^	9.34 × 10^−7^	40	Yes
*SAMD9L*	7	92759368	5.32 × 10^−5^	NA	0	No
** *EZH1* **	17	40852293	6.24 × 10^−5^	3.83 × 10^−3^	23	No
** *MRPS23* **	17	55916287	7.55 × 10^−5^	2.14 × 10^−3^	5	No
** *SIGLEC5* **	19	52114756	8.54 × 10^−5^	9.34 × 10^−7^	20	Yes
*TMED7*	5	114948905	9.56 × 10^−5^	7.53 × 10^−3^	1	No

TWAS, transcriptome-wide association study; Chr, chromosome; BP, start base gene position; *P*_FUSION_, the most significant *p*-value in 44 tissues; *n*, the number of tissues with *P*_FUSION_ < 0.05; NA, *p* > 0.05 in every tissue; Reported, whether reported to be associated with periodontitis before; Bold, significant genes after UTMOST and FUSION analysis.

## Data Availability

Genome-wide summary statistics of the GLIDE consortium are available at: https://data.bris.ac.uk/data/dataset/2j2rqgzedxlq02oqbb4vmycnc2. Genome-wide summary statistics of FINNGEN are available at: https://storage.googleapis.com/finngen-public-data-r7/summary_stats/finngen_R7_K11_PERIODON_CHRON_INCLAVO.gz.

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
