# Peer review of "Genome- and Transcriptome-Wide Association Studies Identify Susceptibility Genes and Pathways for Periodontitis"

_cells, 2022, doi:10.3390/cells12010070_

Round 1
Reviewer 1 Report
In line 47, the authors mentioned that statistical power is a challenge for GWAS. It may be better that the related references solving this issue should be added.
In line 104, please explain why the threshold was used.
In line 151, is the threshold calculated using Bonferroni correction?
In line 161, please explain how to locate the SNPs for 16q21. Are these SNPs all located within 16q21? In GWAS, usually the nearby SNPs (not located in genes but within a fixed distance) can also associate with the corresponding genes. Do you think such SNPs should be included in this case?
In line 188, please explain why SAMD9L was not available in FUSION.
Reviewer 2 Report
This work aims at deepening the molecular and genetic mechanisms underlying the pathogenesis of periodontitis, through a multistep approach. As a first step, a genome-wide meta-analysis allowed the identification of a total of eight single nucleotide polymorphisms (SNPs) associated with LINC02141 gene. Then, the GWAS summary statistics was combined with Genotype-Tissue Expression (GTEx) database of expression quantitative trait loci (eQTL) data, in order to perform a cross- and a single-tissue transcriptome-wide association studies (TWAS). Finally, the expression of the most relevant candidate genes was checked in public datasets. EZH1 gene was significantly upregulated in periodontitis tissue samples, thus defining it a novel candidate for periodontitis, according to the authors’ view.
Although this paper displays some aspects of novelty (e.g, TWAS studies are currently an emerging and successful strategies in studying complex traits and diseases), there are also several limitations and weak points.
INTRODUCTION
Lines 32-34: “Periodontitis, a chronic inflammatory condition resulted from the interaction of bacteria with environmental and host risk factors, can cause the loss of the gingiva, bone, ligament, and even the entire tooth”.
Comment: The authors should provide the correct clinical definition of periodontitis.
Lines 35-37: “The disease is initiated and sustained by bacteria in dental plaque. Its severity is determined by host and environmental risk factors, both modifiable (for instance, oral hygiene) and immutable (for instance, genetic susceptibility). Concordance for periodontitis after adjusting behavioral variables, including smoking, was about 50% in twins, suggesting a worthwhile genetic predilection to the etiology of periodontitis”.
Comment: The authors did not correctly describe the multifactorial origin of periodontitis; in particular, the authors should: i) discuss in more detail the role of genetics in the aetiology of this disease, ii) comment on which and how many genes are known in relation to periodontitis, to date iii) explicit which are the known environmental factors involved and finally, it would be useful to understand what they mean with “host risk factors”.
Lines 43-44. Comment: The authors reported only two genes, GLT6D1 and SIGLEC5, as previously described in relation to periodontitis. Are there other genes that were identified as associated with periodontitis?
Lines 50-52: “Furthermore, strong linkage disequilibrium (LD) within gene-content haplotypes makes it difficult to accurately pinpoint causal variants, target functional genes, and study regulatory machinery”.
Comment: GWAS studies do not identify causal variants.
Lines 69-73: “Using UTMOST and FUSION in combination, Zhu et al. identified two novel susceptibility genes (DCAF16 and CBL) for lung cancer [17]. Cristina et al. used UTMOST to analyze the largest autism spectrum disorder (ASD) GWAS summary statistics and found that NKX2-2 and BLK were associated with ASD but not restricted to the brain tissue”.
Comment: in this section, the authors bring examples regarding the identification of novel susceptibility genes involved in some complex diseases, taking advantage of UTMOST and FUSION methods. The authors dedicated too many lines in describing the software for TWAS, this should be moved to the Methods section.
METHODS
Lines 83-38: “GLIDE is a systematic meta-analysis statistic consisting of nine studies with 17,353 periodontitis cases and 28,210 controls, the specifics of which have been described before [5]. FinnGen comprises of 21,179 periodontitis cases and 287,975 controls”.
Comment: the authors should provide statistical details regarding these two cohorts (e.g, age range, ancestry).
Line 92: “GLIDE and FinnGen were incorporated into a meta-analysis using Metal software”
Comment: the authors should also provide the SCHEME option that they used for the meta-analysis.
Lines 104-105: “The transcriptome-wide significance for the joint test was set up as P-value <1×10–4”.
Comment: the authors should explain why the P-value significant threshold was set up to 1×10–4 (e.g., references).
Lines 115 -122: “GSE10334 [22] and GSE16134 [23] (https://www.ncbi.nlm.nih.gov/geo/) were queried to verify whether candidate genes were differentially expressed in periodontitis sites compared with unaffected sites. In GSE10334, 90 non-smokers with periodontitis provided a total of 247 tissue samples (64 from healthy sites and 183 from periodontitis). RNA was extracted, amplified, reverse-transcribed, labeled, and hybridized using Affymetrix Human Genome U133 Plus 2.0 arrays. GSE16134 contained 120 systemically healthy non- 120 smokers with periodontitis, including 310 gingival tissue samples (241 samples from affected and 69 from unaffected sites)”.
Comment: The GSE16134 is a database that collects data on the association between subgingival bacterial profiles and gene expression patterns in gingival tissues of patients with periodontitis.
The dataset could be enriched with only the genes whose expression correlates with specific bacterial profiles. The authors should properly discuss the reason they chose this dataset.
RESULTS
Lines 152-153: “All of these SNPs were non-coding variants situated in introns of lncRNA LINC02141 and were in high LD”.
Comment: Since only eight SNPs with a significative pvalue were detected, and the authors did not deepen the role that this lncRNA could play in periodontitis’ pathogenesis, this result does not add any value to the manuscript.
Lines 170-171: “It is noteworthy that SIGLEC14 passed the Bonferroni correction threshold of 2.89×10–6 (0.05/17,290)”.
Comment: SIGLEC14 is the only gene that passed the Bonferroni correction; the authors did not discuss whether this gene could have a role in periodontitis, this result does not add any value to the manuscript. Furthermore, since the other genes did not pass the Bonferroni correction, the authors should properly comment on this aspect.
Line 165-180: In this paragraph, the authors describe that they performed a cross- and a single-tissue TWAS considering a broad range of different human tissues. What’s the point of considering all this broad range of tissues, if the authors are studying an oral disease, and the tissues relevant to the disease are absent? The only tissue that could be related to periodontitis is the “Minor_Salivary_Gland”, which was never discussed.
Line 165-169. Comment: the authors should provide the complete significative results obtained with UTMOST and FUSION software (at least, as supplementary tables).
Lines 194-195: “TWAS is widely used to infer disease-related genes based on SNP genotypes. Therefore, if the genes calculated by TWAS are actual, their expression should be dysregulated in periodontitis. Because the whole-genome microarrays did not include SIGLEC14, its 196 differential expression was difficult to evaluate according to GSE10334 and GSE16143”.
Comment: The words “calculated”, “actual, and “difficult” are not appropriate in this context.
Lines 194, 199: GSE16143 refers to a database collecting transcriptional regulatory genes in secondary wall biosynthesis in Arabidopsis thaliana.
Line 207 (all paragraph “Functional annotation of EZH1”)
Comment: The authors focused on the EZH1 gene without giving a clear explanation of this choice. The reader could guess that maybe it is due by the fact that it is the only gene whose expression is increased in both the GEO datasets, but the author did not comment about this.
Lines 208-209: “EZH1 could result in inflammation by interacting with 41 chemicals including smoke, soot and so on, with an inference score of 140.7 (Figure 5a, b)”.
Comment: What did the authors mean by “EZH1 could result in inflammation”?
Furthermore, the authors should properly support this finding with literature references regarding the association between periodontitis and environmental agents, such as smoke and soot.
Lines 212-217: “The robust 212 statistical evidence for GO enrichment was observed for cell-substrate junctions (GO: 213 0030055, Padjust =1.04E-11) and focal adhesion (GO: 0005925, Padjust =1.04E-11) (Figure 5c, Ta- 214 ble S4). KEGG analysis revealed that EZH1 expression was closely associated with Epstein-Barr virus (EBV) infection, lysosomes, and protein processing in the endoplasmic 216 reticulum, all of which are associated with periodontitis (Figure 5d, Table S4)”.
Comment: the authors should provide adequate and proper references to support this finding, also in the discussion section. In particular, the authors should clarify:
- the link between Epstein-Barr virus (EBV) infection (to my knowledge, no relation between EBV and periodontitis is known).
- the role of lysosomes, and protein processing in the endoplasmic reticulum with periodontitis.
DISCUSSION
Line 226: “Chromosome 16q21 is a susceptibility locus for periodontitis”.
Comment: the meta-analysis detected only eight SNPs within in introns of lncRNA LINC02141; this is not sufficient to conclude that chromosome 16q21 is a susceptibility locus for periodontitis.
Lines 226-228: “UTMOST then served as a cross-tissue TWAS method and identified six susceptibility genes associated with periodontitis, four of which were validated by FUSION”.
Comment: the authors identified six susceptibility genes associated with periodontitis, but the analysis was not carried out in specific tissues relevant to the disease they are studying. The only tissue that could be related to periodontitis is the “Minor_Salivary_Gland”, which was never discussed. For this reason, this is not sufficient to conclude that these are susceptibility genes for periodontitis.
Line 234: “In 16q21, there were multiple SNPs in the introns of lncRNA LINC02141 with P <5e-08”.
Comment: the authors identified only eight SNPs. “Multiple” is not appropriate.
Lines 242-243: “However, only association between the SNPs in 16q21 and periodontitis was observed”.
Comment: The author failed to support and discuss this finding properly.
Lines 245-247: “The integration of GWAS and eQTL data has been successfully used to locate causal genes. Li et al. used a Bayesian statistical method called Sherlock to systematically incorporate GWAS and eQTL data for periodontitis from whole blood”.
Comment: TWAS do not allow the identification of causal genes.
Lines 252-252: SIGLEC5 and SIGLEC14 were certified, and EZH1 and MRPS23 were newly identified, proving that TWAS is a powerful strategy for studying periodontitis etiology.
Comment: What did the authors mean by “certified”? Further, since there were several weak points in this work, the authors should not conclude that TWAS is a powerful strategy for studying periodontitis’ etiology.
Line 231: “Genes co-expressed with EZH1 were considerably enriched in biological processes like cell-substrate junctions”.
Comment: The authors should properly comment on this finding in relation to periodontitis.
Lines 261-267: “Meanwhile, Ezh1 is the most upregulated histone methyltransferase during dendritic cell maturation, silencing of which significantly suppressed Toll-like receptors triggered production of cytokines, including IL-6, TNF-α, and IFN-β, in dendritic cells (DCs) 263 and macrophages [34].”.
Comment: If the authors refer to the protein encoded by the EZH1 gene, Ezh1 should not be written in italics. Further, if they are not referring specifically to the murine protein, it must be written in capital letters.
Lines 261-267: “When infected with Porphyromonas gingivalis, a major pathogen of periodontitis, both cell-matrix and cell-cell adhesion interfered in the oral keratinocyte cell line, which could compromise the integrity of the epithelia and result in the generation of gingival pockets between the cementum and periodontal epithelium.”
Comment: “Porphyromonas gingivalis” must be written in italics. Further, the authors should better describe the association between periondontal bacterial and periodontitis development in the introduction section.
Line 278-283: “Meanwhile, lysosomes are essential for autophagy, fusing with autophagosomes to form autolysosomes [40]. When periodontal disease develops, autophagy has an impact on immunological function, inflammation, and the homeostasis of alveolar bone [41]. Altogether, the GO and KEGG enrichment analysis results provide additional evidence for the potential role of EZH1 in the pathological process of periodontitis”
Comment: The authors did not correctly discuss and support these findings in relation to periodontitis.
Line 287: “eQTL information from GTEx was employed in TWAS analysis without gingival-related cells or tissue”.
Comment: This is a strong limitation of the study. The authors must properly discuss this weak point. The only tissue that could be related to periodontitis is the “Minor_Salivary_Gland”, which was never discussed.
Furthermore, in the manuscript there are the following typing errors:
- Line 19: “Risk genes were evaluated to determine if they were deferentially expressed in periodontitis sites”.
- Line 64: “as a kidney disease risk gen”.
- Line 101: “were then performed by inputting GWAS periodontitis summary statistics”.

Reviewer 3 Report
The manuscript is focused on the genome and transcriptome association study for periodontitis. Genome-wide meta-analysis of 38,532 patients and 316,185 healthy controls was per- 16 formed. Then, cross- and single-tissue transcriptome-wide association studies (TWAS) were con- 17 ducted based on GWAS summary statistics and the Genotype-Tissue Expression (GTEx) project. Authors found that EZH1 was identified as a novel susceptibility gene for periodontitis by TWAS and 23 was significantly upregulated in periodontitis-affected gingival tissues. Data are clear and enough to publish to Cells.
Round 2
Reviewer 2 Report
For the authors
Overall, the manuscript has been revised taking into consideration the Reviewers’ comments and requests. However, there are still some weak points to be clarified, especially in the Introduction and in the Methods section.
INTRODUCTION
1. Lines 32-34: “Periodontitis, a chronic inflammatory condition resulted from the interaction of bacteria with environmental and host risk factors, can cause the loss of the gingiva, bone, ligament, and even the entire tooth”.
Comment: The authors should provide the correct clinical definition of periodontitis.
Response: Thank you for your valuable suggestion. We have provided the clinical definition of periodontitis in lines 32-35 [1,2], referring to the clinical definition of periodontitis developed by the 2017 World Workshop on the Classification of Periodontal and Peri-Implant Diseases and Conditions.
Lines 32-35: “Periodontitis is a microbially-associated, host-mediated inflammation characterized by the loss of periodontal tissue support manifested by clinical attachment loss (CAL) and radiographically assessed alveolar bone loss, presence of periodontal pocketing and gingival bleeding.”
References
1. Tonetti, M.S.; Greenwell, H.; Kornman, K.S. Staging and grading of periodontitis: Framework and proposal of a new classification and case definition. J. Clin. Periodontol. 2018, 45 Suppl 20, S149-S161, doi: 10.1111/jcpe.12945.
2. Papapanou, P.N.; Sanz, M.; Buduneli, N.; Dietrich, T.; Feres, M.; Fine, D.H.; Flemmig, T.F.; Garcia, R.; Giannobile, W.V.; Graziani, F.; et al. Periodontitis: Consensus report of workgroup 2 of the 2017 World Workshop on the Classification of Periodontal and Peri-Implant Diseases and Conditions. J. Periodontol. 2018, 89 Suppl 1, S173-S182, doi: 10.1002/JPER.17-0721.
Comment_revision2: The authors should explicit that, from a genetic point of view, periodontitis is a multifactorial disease; in this way, it would be easier for the reader to follow the thread of the next section, regarding the impact of genetic variations on periodontitis’ development.
2. Lines 35-37: “The disease is initiated and sustained by bacteria in dental plaque. Its severity is determined by host and environmental risk factors, both modifiable (for instance, oral hygiene) and immutable (for instance, genetic susceptibility). Concordance for periodontitis after adjusting behavioral variables, including smoking, was about 50% in twins, suggesting a worthwhile genetic predilection to the etiology of periodontitis”.
Comment: The authors did not correctly describe the multifactorial origin of periodontitis; in particular, the authors should: i) discuss in more detail the role of genetics in the aetiology of this disease, ii) comment on which and how many genes are known in relation to periodontitis, to date iii) explicit which are the known environmental factors involved and finally, it would be useful to understand what they mean with “host risk factors”.
Response: Thank you for your valuable suggestion. We have described the multifactorial origin of periodontitis according to your requirements in lines 39-51.
Lines 39-51: “Genetic variants may considerably contribute to the development of periodontitis because they may change immune responses and define individual disparities. Periodontitis had approximately 50% heritability after adjusting for behavioral variables in a twin study, suggesting a worthwhile genetic predilection to the etiology of periodontitis. A systematic review revealed that twenty-nine per cent of the variance of periodontitis traits is related to genetic factors based on twin and family studies. Variations in at least 65 genes, especially genes involved in inflammation, have been implicated to be associated with periodontitis. Besides the genetic factors, the negative impacts of environmental risk have been acknowledged for decades. Tobacco use is one of the most significant avoidable risk factors for the occurrence and development of periodontal disease. Moreover, medicines, poverty, obesity, uncontrolled diabetes and mental stress are also strongly associated with periodontitis. Thus, the severity of periodontitis is determined by host and environmental risk factors, both immutable and modifiable.”
Comment_revision2:
Lines 40-41:“Genetic variants may considerably contribute to the development of periodontitis because they may change immune responses and define individual disparities”.
In which way genetic variants could impact on immune responses and individual disparities? Are there any published studies that are currently providing evidence of an association between genetic variants and immune response/individual disparities in periodontitis?
Lines 51-52: “Thus, the severity of periodontitis is determined by host and environmental risk factors, both immutable and modifiable.”
It is still not clear what the authors mean by “host risk factors both immutable and modifiable”.
3. Lines 50-52: “Furthermore, strong linkage disequilibrium (LD) within gene-content haplotypes makes it difficult to accurately pinpoint causal variants, target functional genes, and study regulatory machinery”.
Comment: GWAS studies do not identify causal variants.
Response: Thanks for your valuable advice and we agree. We have simplified our expression in lines 62-64.
Lines 62-64: “Furthermore, strong linkage disequilibrium (LD) within gene-content haplotypes makes it difficult to identify causal variants and target functional genes.”
Comment: The terms “causal variants” should not be used for GWAS studies results.
4. Lines 69-73: “Using UTMOST and FUSION in combination, Zhu et al. identified two novel susceptibility genes (DCAF16 and CBL) for lung cancer [17]. Cristina et al. used UTMOST to analyze the largest autism spectrum disorder (ASD) GWAS summary statistics and found that NKX2-2 and BLK were associated with ASD but not restricted to the brain tissue”.
Comment: in this section, the authors bring examples regarding the identification of novel susceptibility genes involved in some complex diseases, taking advantage of UTMOST and FUSION methods. The authors dedicated too many lines in describing the software for TWAS, this should be moved to the Methods section.
Response: Thank you for pointing it out. Now we have moved them to the Methods section in lines 102-104 and lines 114-115.
Lines 102-104: “Considering a certain number of relevant tissues were scarce and the similarity in transcription regulation across multiple tissues, researchers created a cross-tissue analysis method called UTMOST (Unified Test for MOlecular SignaTures).”
Lines 114-115: “Some methods train independent imputation models for various tissues due to the tissue transcription specificity.”
Comment_revision2: The authors did not modify this section as requested. In particular, lines 104-106 should be moved in the Introduction section.
Lines 116-117: This sentence is not so clear. Which are these “methods” mentioned by the authors? Further, this text edit is not coherent with my previous comment.
